# Surveillance of Wildlife Viruses: Insights from South Australia’s Monitoring of Rabbit Haemorrhagic Disease Virus (RHDV GI.1 and GI.2)

**DOI:** 10.3390/v16101553

**Published:** 2024-09-30

**Authors:** David E. Peacock, Amy Iannella, Ron G. Sinclair, John Kovaliski

**Affiliations:** 1Davies Livestock Research Centre, School of Animal and Veterinary Sciences, University of Adelaide, Roseworthy, SA 5371, Australia; 2Foundation for Rabbit Free Australia, P.O. Box 145, Collinswood, Adelaide, SA 5081, Australia; 3School of Animal and Veterinary Sciences, University of Adelaide, Roseworthy, SA 5371, Australia; 4Independent Researcher, 6/43B Bridge Street Kensington, Kensington, SA 5068, Australia

**Keywords:** RHDV2, lagomorph, *Oryctolagus cuniculus*, Turretfield, monitoring, vectors, ecology, burrow searching, citizen science, Australia

## Abstract

Surveillance of wildlife virus impacts can be passive or active. Both approaches have their strengths and weaknesses, especially regarding cost and knowledge that can be gained. Monitoring of rabbit haemorrhagic disease virus (GI.1 and GI.2) in South Australia has utilised both strategies and their methods and gained insights are discussed. Active strategies to monitor the continuing impact of rabbit haemorrhagic disease virus 2 (GI.2) on susceptible lagomorphs in countries such as the USA, Mexico, South Africa, Spain, France and Portugal are encouraged to gain critical insights into the evolution, spread and impact of this virus. Furthermore, there are lessons here for the international monitoring of diseases in wildlife, particularly where there is a risk of them becoming zoonotic.

## 1. Introduction

Monitoring the evolution and spread of viral diseases is critical for managing and assessing their impacts on wildlife, livestock and pest animals, particularly where there is a risk that these diseases are, or may become, zoonotic [1]. Recent examples of monitoring the evolution and spread of viral diseases include avian influenza [2], African swine fever [3] and rabbit haemorrhagic disease virus (RHDV; [4]). They have been monitored using passive and/or active methodologies. Passive monitoring of outbreaks relies on opportunistic tissue collection, and spontaneous public reports on disease spread which may lead to tissue samples and confirmation of disease. In contrast, active monitoring involves regular activities at targeted sites by assigned personnel. These regular activities result in epidemiological data reporting on both the presence and absence of disease, and population dynamics through capture–mark–recapture studies and vector screening [5].

Rabbit haemorrhagic disease virus (RHDV1/GI.1 [6]) became established in Australia in late 1995 [7]. The initial outbreak of RHD varied in its impact with up to a 95% reduction in pest rabbit (*Oryctolagus cuniculus*) populations [8], and the greatest impacts were generally recorded in more arid regions [9,10]. Its spread, impact, evolution and developing resistance in rabbits have been monitored by a variety of both active and passive programs. Passive monitoring relied on information from the public but also regionally based government officers, television, newspaper and radio media, local veterinarians and other researchers. Active surveillance was led initially by the Commonwealth government funding national monitoring sites across the country at which ‘intensive’ active surveillance took place. In South Australia, active surveillance of the virus and of rabbits has utilised capture–mark–recapture techniques, shot sample transects, carcass searches and fly vector screening. Now, as the spread of RHDV2/GI.2 across the world raises concerns [11], including its impact on lagomorphs in the USA [12], Mexico and South Africa [13], the South Australian RHDV monitoring experience can provide insights relevant to gaining a better understanding of the impacts of RHDV2 and other wildlife pathogen monitoring systems globally.

Passive monitoring of wildlife disease is often implemented because it is seen as a relatively fast, simple and cheap option. However, our experience with spontaneous landholder/public reporting was that it sometimes provided unreliable, patchy or sporadic data that gave little more than a rough idea of outbreak frequency and geographic distribution. This is of limited use for identifying epidemiological changes and impacts. Conversely, while active monitoring approaches are much more informative, they also require substantial investment, so should be chosen carefully based on the questions to be addressed. A mixture of active and passive monitoring can provide complementary data, facilitating insights into both wide-scale outbreak activity and detailed demographic, epidemiological and evolutionary changes. In this paper, we share our experience with five different wildlife virus surveillance techniques and summarise their relative advantages and disadvantages to facilitate the development of considered monitoring strategies.

## 2. Materials and Methods

### 2.1. Passive Monitoring

Passive surveillance in South Australia during the initial spread of RHDV1/GI.1 (1995–1997) [7] was promoted by both frenzied media interest, given the virus’s escape from quarantine on Wardang Island was classified as an exotic disease outbreak, and through the cultivation of long-term relationships between our state government research and advisory staff with landholders and veterinarians. Passive surveillance of rabbits across South Australia has resulted in the detection and provision of hundreds of samples from dead rabbits. This notably enabled monitoring of the initial spread [7] and subsequently, the seasonal reoccurrence of RHDV1/GI.1. These samples importantly contributed to later genetic studies and understanding the rapid evolution of the virus [14].

Following that initial mainland spread, we used the media (radio, television and newspapers) to promote the message that the virus offered Australia the opportunity to have an important impact on wild rabbits, regarded as the nation’s number one pest animal. Once the official release of the virus was sanctioned some 18 months after the initial escape, we used the media to again spark interest in the community to report dead rabbits and this form of short-term passive monitoring resulted once again in rabbit samples being obtained for testing.

### 2.2. Active Monitoring: Capture-Mark-Recapture Studies with Serology

In October 1996, the South Australian government invoked its Biological Control Act allowing the official and intentional release of RHDV1 to help control wild rabbits. The first official release occurred at the Turretfield Agricultural Research Centre on 22 October 1996 where 19 wild rabbits were injected with the virus and released in a small area of the Research Centre rarely used for agricultural purposes. Site details are given in Peacock and Sinclair (2009) [15]. Two days after the release, the site was inspected and an unexpectedly high number of dead rabbits was found. The site was revisited to search for carcasses several times over the next 10 days, by which time no new dead rabbits were located. A month later, cage traps were taken to the site to catch (and begin tagging) rabbits and obtain blood samples to determine if the remaining rabbits had antibodies to RHDV1. This was the first of 153 trapping sessions at roughly 8-week intervals over the next 26 years. Data collected on trapped rabbits included weight, sex, reproductive status, presence of clinical myxomatosis, rabbit flea abundance and if a blood sample was obtained. The rabbits were ear-tagged (a unique number for each animal including kittens down to 100 g live weight) and released at the warren on which they had been trapped (details given in [15]). Blood samples were analysed for RHD and myxomatosis antibodies as per [16,17,18,19]. Traditional rabbit survey methods (such as spotlight transect counts) were impracticable at Turretfield because of the difficult and rocky terrain. We relied on minimum known-to-be-alive (KTBA) data to reflect rabbit abundance and provide a visual picture of the impact of RHDV and myxomatosis. To age trapped rabbits, we used an average growth rate of 10 g/day and for rabbits over 1300 g, this provided a minimum age [15]. A rabbit was assumed to be alive at every trapping trip between its first capture or estimated birth date and the last time it was trapped or found freshly dead.

### 2.3. Active Monitoring: Carcass Sampling

Carcass sampling involves visual searches for dead rabbits at the site of a known population, including smelling of burrows and visual inspection of burrows using a torch (Figure 1). For each rabbit carcass recovered, necropsy is conducted on site and any visual signs of RHDV or myxomatosis and timing of death are noted. These include pale reticulated liver and/or dark enlarged spleen common in RHDV mortalities, lesions and excretions around the eyes and genitals for myxomatosis, as well as the presence and size of maggots for estimating date of death [20]. Where carcasses are fresh, the liver is preferentially sampled (a known high source of virus [21]), followed by other remaining organs. For older, significantly scavenged or decomposed carcasses, bone marrow can be extracted from leg bones to attempt viral RNA extraction. New gloves and a scalpel are used for each carcass to minimise the risk of cross contamination. Carcasses are left where they were found to continue spreading the virus naturally. The tissues are subject to RT-PCR analysis to confirm the presence of RHDV, identify genetic variants of the pathogen and allow pathogen evolution to be traced over time [22].

During the initial spread of RHDV1 (GI.1) [7] its impact on rabbit populations was assessed in November 1995 through a count of carcasses on the surface of 22 plots in the Ikara–Flinders Ranges National Park, South Australia.

### 2.4. Active Monitoring: Spotlight Transects and Shot Samples

In 1996, immediately after the establishment of RHDV1/GI.1 in Australia, national monitoring programs were established, including the ‘Rabbit Calicivirus Disease Monitoring and Surveillance Program’ and the ‘Epidemiology Research Program’ [23,24]. These programs involved Australian states and territories nominating sites to be actively monitored to assess RHDV1/GI.1 activity and its ecological impacts. In South Australia, the two sites were a coastal temperate site in and adjacent to the Coorong National Park, and a semi-arid site in the Ikara–Flinders Ranges National Park. At these sites, monthly spotlight transects were conducted using a standard protocol to provide an index of abundance. These counts tracked the changes in rabbit and native herbivore abundance [10,25]. Other commonly used census options include dung quadrats [26] and remote cameras.

In addition, attempts were made to collect a shot sample of 20 rabbits on adjacent properties. Shot samples can also be obtained from licensed hunters. This technique is particularly applicable in Australia where the rabbit is an introduced pest species, though we recognise that for many other lagomorph species, destructive sampling is not appropriate.

### 2.5. Active Monitoring: Vector Surveys

As well as targeting host populations directly, the evolution, prevalence and spread of wildlife pathogens can be monitored through vectors. Vector-based monitoring [20] has been used internationally for surveillance of mosquito-borne viruses, such as West Nile Virus, Ross River and Barmah Forest Viruses [21], and for Blue-tongue Virus in midges [22], but has typically been overlooked for fly-borne diseases such as RHDV. In 1985, Vogt et al. [23] described a simple, cost-effective, wind-oriented fly trap, which is capable of trapping large numbers of several carrion- and manure-visiting fly species. In 1998, Asgari et al. [24] used these traps to collect RHDV-carrying flies and were able to detect the presence of the virus in flies using RT-PCR both during an epizootic and in the month prior to and after rabbits killed by the virus were found. Although their study effectively demonstrated the utility of fly traps for RHDV detection, the traps have not been widely adopted for monitoring purposes.

During 2013 and 2014, we conducted a pilot fly surveillance study across five field sites in South Australia’s Gawler/Barossa region surrounding our Turretfield study site. Wind-orienting fly traps (pictured in Figure 2) were constructed as described by Vogt et al. [23]. One fly trap was established at each of several field sites and baited with a slurry of minced liver, cattle or sheep dung, sodium sulphide (Na2S) and water. The trap chamber where flies accumulated in each trap was replaced approximately weekly throughout the expected late winter/spring RHDV1 outbreak season (August–November) and the bait was rehydrated with water and topped up with minced liver. Trapped flies were identified, counted, and frozen at −80 °C until RNA extraction. Carcass searches were also performed on each visit as described under “3. Carcass Sampling” above. No attempt was made to collect flies directly from carcasses.

RHDV VP60 capsid RNA was amplified using the primers RHDVf4846, RHDVr6059, RHDVf5926 and RHDVr6986 [25]), sequenced from both carcasses and pooled fly samples at each site and date, then de novo assembled to identify all RHDV variants present. The variants were then screened for recombination events and their relationships were visualised through phylogenetic analysis. Full genetic methods are detailed in Appendix A [24,25,26,27,28,29,30,31,32,33,34,35,36,37].

## 3. Results

### 3.1. Passive Monitoring

Knowledge gained through passive surveillance by the public can be limited by their observational skills and awareness of the relevance of observations [38]. So, to be useful, an intentional public recruitment process needs to go with a relevant education program that gives the public guidance on what information is required as well as accessible places to lodge reports [39]. For example, action by a member of the public, triggered by information in the media, lead to confirmation of the initial mainland long-distance spread of RHDV1/GI.1 at Yunta c. 300 kms north-east of Wardang Island (SA) from where the virus had escaped pen trials [40,41]. The ramification of this discovery was a national decision that attempts to contain this exotic disease were now unlikely to be successful, stopping all work underway to achieve this. However, several incidental discussions over a year later indicated that the virus had likely impacted rabbits in a plume away from Wardang Island at sites over 175 kms apart. A landholder from a property near Lake Torrens c. 180 kms north-west of Yunta regularly counted rabbits on a set piece of road and noted his counts dropped from well over a hundred to almost zero over a few days. He also noted large numbers of scavenging black kites (*Milvus migrans*) feeding on dead rabbits in the sandhills on his property. These observations were not formally reported. National Parks and Wildlife field staff carrying out vegetation surveys on a property c. 70 kms north-east of Yunta recorded in their notebooks the presence of large numbers of dead rabbits on warrens and noted it was obviously not myxomatosis, again not formally reported. At approximately the same date, professional rabbit shooters on a property 35 kms west of the confirmed Yunta RHDV1/GI.1 samples saw, but did not report, many dead rabbits. Thus, of four known observations of dead rabbits in that region of the initial mainland outbreak, only one was officially reported and led to the confirmation that the virus had spread.

Within our state government organisation, a stable core of scientific staff over 20–30 years maintained relationships with individuals including landholders, regional officers and veterinarians. These core staff provided a personal point of contact for sample submissions that was stable across time. We believe that these long-term relationships and networks introduced a personal touch which included information about local rabbit numbers and feedback to the community on whether samples submitted were positive for RHDV and this helped to maintain public interest in reporting outbreaks over time. An example of where we see the critical value of these long-term relationships was when RHDV2/GI.2 was detected in New South Wales and then rapidly spread across Australia. Our passive monitoring network provided samples from 108 dead rabbits in South Australia, twice the number obtained in the much more densely populated New South Wales [42,43,44]. It also provided our first, and only one of five, RHDV2/GI.2 infected and killed European brown hares [45]. To achieve wide participation we found that passive sampling needs to be easy for the landholders and community with clear instructions [46], also noted in an Australian fox tissue sampling study which successfully procured 3122 samples [47]. The Australian release of the RHDV1 K5 variant provided clear guidelines for community passive sampling and the online RabbitScan application (www.rabbitscan.org.au (accessed 30 September 2024)) for submission of information [39]. Alongside active monitoring of some experimental sites, this data showed minimal impact of this virus variant [38,48]. This passive sampling is easier in the initial disease impact period or for unique virus releases than for long-term passive monitoring research, and is also easier in higher populated areas than in remote regions, though the rare remote samples can provide powerful insights—for example, on myxoma viral evolution and spatial distribution [49]. Passive sampling can however give a biased picture due to the primary provision of positive samples. Despite passive surveillance being useful for tracing the initial viral spread and contributing some tissue for genetic data to study RHDV evolution [25], it did not provide critical epidemiological information required to assess the impact of the virus over time or to inform management decisions.

### 3.2. Active Monitoring: Capture–Mark–Recapture Studies with Serology

The most informative form of active surveillance for us has been through capture–mark–recapture study of a host population, coupled with serological testing at each capture. Serology is a critical tool to understand if and when a virus is circulating in a host population, particularly if carcasses in which the virus might be identified are difficult to find. Regular serological testing can reveal who has been infected or challenged, seasonal patterns to virus activity, interactions between different circulating viruses and how impactful prior exposure is on subsequent virus challenges. While labour intensive, these studies can reliably assess outbreak frequency, morbidity and mortality rates and population impacts [20,50,51]. When maintained long-term, capture–mark–recapture studies can reveal epidemiological changes over time, and provide insights into the evolution and adaptation of both host and pathogen [52,53,54,55,56].

In total, between 1996 and 2022, we had 10,351 live captures of 5216 individual rabbits from which 9759 blood samples were collected for antibody testing. Figure 3 shows that in the first 10 years, RHDV1 outbreaks (as determined by the finding of fresh carcasses and changes in survivors’ serology) occurred mostly every second year in late winter/early spring [50]. From 2006 onwards RHDV1 outbreaks occurred every year until 2016 when RHDV2 appeared and the timing of outbreaks changed to late autumn/early winter. In addition, from 2009 to 2011 the population increased after the outbreak in those years reflecting the development of resistance in the rabbit population [54,56,57]. The low population size between 2003 and 2006 was likely due to persistent virulent myxomatosis during that time.

We know that the KTBA data only includes the trappable portion of the population, a standard premise of all such activities, but we believe we were able to capture a high proportion of rabbits present in most years other than in 1999 because when RHD outbreaks occurred, most carcasses found had ear tags. In 1999 the population was high and we had trap saturation. When the RHD outbreak occurred, only 50% of carcasses were found to have ear tags suggesting that the population might have been double that shown in Figure 3. Our capture–mark–recapture data has been used in a number of population models [52,53,58] but only in Barnett et al. [52] did the model closely follow the pattern seen in Figure 3.

The first rabbits on the site confirmed to have RHDV2 were in 2016 and our serological data gave evidence that RHDV2 could overcome antibodies to RHDV1 [59]. Since then, the population has continued to decline with RHDV2 remaining as the dominant virus [51]. Concomitant with the arrival of RHDV2, there has been a change in the behaviour of the rabbits, particularly in adults. Their trappability dropped off markedly. Adults must have been present in warrens where we caught kittens, but only occasionally did we catch an adult there, and some adults went two or more years between captures. It might be because with so few rabbits on the site there might have been behaviour changes reducing trappability, or fewer rabbits meant more food and a lower attractiveness of trap bait, but really we are at a loss as to why adults became so hard to trap when the only change we are aware of is the replacement of RHDV1 by RHDV2.

The combination of data from our long-term capture–mark–recapture work combined with RHD and myxomatosis antibody serology and virus sampling from carcasses has allowed exploration of changes in population dynamics, epidemiology, the interaction between RHD and myxomatosis, rabbit genetics and host–virus co-evolution reported in 25 publications. Temporal analysis of the 15 RHDV1 outbreaks in the first 19 years showed changes in outbreak frequency (Figure 3). Similarly, a spatial analysis of these outbreaks is proposed as per Pfeiffer et al. [60] to test for any inter-annual outbreak patterns. Our capture–mark–recapture research at Turretfield has provided critical data on age-specific mortality, showing that 82% of young rabbits do not live past 6 months, only 8% reach 12 months of age (see Figure 4), and the longest-lived rabbits were all females [61,62]. Serological samples also gave us an insight into other rabbit diseases such as myxomatosis and its interaction with RHD [52]. It showed that the long-proposed idea that myxomatosis was no longer as effective in controlling rabbit numbers was incorrect. Long-term serum collection determined that myxomatosis was not only effective but its outbreak occurrence and prevalence had changed [58].

### 3.3. Active Monitoring: Carcass Sampling

Rabbit warrens at Ikara–Flinders Ranges National Park during the initial spread of RHDV1 (G1.1) were assessed to be at an average density of 50 warrens/km^2^, with 10 to 30 rabbits per warren. Carcass counts on 22 plots at seven representative sites in this Park yielded 9.6 ± 10.8 carcasses/ha, indicating in that month there were about 1 million dead rabbits on the surface of the park, and therefore RHDV1 (GI.1) had just killed around 30 million rabbits in South Australia [41].

Carcass searches are particularly valuable when performed routinely at a capture–mark–recapture study site because carcasses are frequently tagged and their individual demographic and serological history is known. This combined data can yield valuable epidemiological insights, as demonstrated at our Turretfield site. Routine carcass searches between 1996 and 2022, generally weekly during late winter/spring until a RHDV1;GI.1 outbreak was detected and similarly during late autumn/early winter for RHDV2;GI.2, yielded 577 largely intact dead rabbits (414 tagged, 163 untagged), 169 remains of dead rabbits (107 tagged, 62 untagged) and 902 loose ear tags (not a guaranteed sign of death but when the breeding season begins these are frequently kicked out of burrows alongside any other debris from long-dead rabbits).

The combination of serology and carcass monitoring at Turretfield enabled assessment of the following:
Infectivity of rabbits:
○In October 1999 active monitoring of Turretfield rabbit warren burrows detected a dead, unfurred 90 g kitten (photo available) in which PCR analysis of its liver sample detected RHDV1; GI.1.Timing of outbreaks:
○The weighing, or estimation of age, of all rabbit carcasses found during RHDV1 outbreaks enabled the assessment of a change in timing and duration of RHDV outbreaks through increased infectivity of juvenile rabbits [50].Assessment of viral variants:
○PCR analysis of virus present in flies and rabbit carcasses revealed that multiple variants of RHDV can be active in an area but generally, one variant is detected in most of the dead rabbits [63].Estimates of mortality:
○RHDV antibody status of animals known to be alive at the trapping session immediately prior to an outbreak combined with carcass collection and serology of survivors identified at the post-outbreak trapping session was used to compare the mortality of rabbits infected during each RHD outbreak and ranged from 45 to 80% (Figure 5) [41].

Physical carcass searches are labour intensive and reliant on the observational capabilities and experience of searchers who are familiar with local rabbit habitats. Where rabbits are warren-dwelling, carcasses deep in burrows identified only by smell, flight paths of carcass-feeding flies or trails of maggots or meat-eating ants may need to be dug out. Not all rabbits die above ground [41], and these can often be quickly taken by scavengers, making visual searches unreliable where rabbit densities are low, or rabbits are inhabiting difficult terrain.

### 3.4. Active Monitoring: Spotlight Transects and Shot Samples

Rabbit shot samples provided liver, blood and eye samples for ageing rabbits, and morphometric measurements to assess virus activity and variant, RHD and myxoma antibody titres and rabbit breeding activity [64]. Changes in abundance from spotlight transects and changes in antibody titres were considered indicative of virus activity and impact [41,51,65].

### 3.5. Active Monitoring: Vector Surveys

The fly traps were highly effective, catching an average of 605 flies per trap each trip. Carcasses were found on only 17 out of 70 search occasions (24%) whereas RHDV was detected more frequently using PCR in 49 out of 70 (70%) fly samples (McNemar’s chi-squared test χ2 = 30.118, df = 1, *p* < 0.001). Hall et al. [26] followed our fly-trapping pilot by optimising the laboratory methods with RT-qPCR and RT-PCR to detect RHDV variants and had similar success with detection of the virus in flies over carcasses. We detected RHDV on flies at all five sites, including those at which no rabbit carcasses were ever discovered. Where the density of rabbits was low no carcasses were found. We know that rabbit populations at low density have higher proportional survival during outbreaks of RHDV [66], limiting the pool of available carcasses. Additionally, carcasses can be well camouflaged, located deep within warrens, or removed by scavengers, so that the number detected is likely to be substantially smaller than the total number of rabbit deaths.

We detected RHDV1 in flies up to six weeks before, as well as after, the discovery of rabbit carcasses, although not on every trip nor at all times when it would have been expected. For example, at one site a fresh rabbit carcass was found on 28 September 2013, and flies tested positive for RHDV in the preceding and following weeks, but a negative result was obtained from flies collected on the same day. The apparent inability to detect RHDV1 in some fly samples, where the virus might reasonably be expected, raises the possibility of false negative results which might be obtained if only a few flies were carrying the virus and none were trapped [24]. Inclusion of replicate fly traps in future monitoring projects will further reduce the chance of false negative results, as well as allowing detection probability to be calculated, providing a measure of confidence in virus presence/absence.

As well as improving detection rate over carcass-searches, vector-based monitoring was far more efficient. Continuous fly-trap monitoring required less than 10 min per site to collect and reset the trap, compared with up to two hours to search for rabbit carcasses at each site. Fly trapping requires no prior experience or specific skill, unlike carcass searches which rely on the searcher’s experience, observational skill and site familiarity. The simplicity of fly trapping lends itself to the use of citizen science for the establishment of extensive disease monitoring systems, both for RHDV and other fly-borne viruses. A community volunteer could freeze flies prior to transport to a processing laboratory. The fly traps are cheap, reusable and durable—our traps remained serviceable after three years of continuous use. However, manual carcass searches remain relevant for use during smaller-scale intensive studies of outbreak epidemiology (such as [50,59]) where demographic details of rabbit mortality are required.

Next-generation sequencing of fly-borne RHDV1 confirmed multiple co-circulating variants at a localised scale (up to 52 unique contigs per pooled fly sample, with a mean of 7.9 contigs), facilitating frequent recombination events throughout an outbreak season (16 recombination events were detected, affecting 51 contigs of 85 total). The importance of recombination in the evolution of RHDV has been postulated previously following the detection of recombination both within [25,67,68] and between [44,69,70] genotypes at national and international scales. Indeed, the RHDV2 genotype (GI.2) which is currently dominant worldwide appears to be a product of recombination which has resulted in increased infectivity in young rabbits and hares [71]. The detection of 16 probable recombination events here evidences the frequency of recombination even among local variants, adding weight to the importance of recombination for producing variation in this virus upon which selection can act.

Phylogenetic analysis grouped the RHDV sequences into three clades. The clade present in a rabbit carcass was not always detected in flies from the same sample site and date. All rabbit carcasses within a year at any given site had the same RHDV1 cluster present, suggesting that local outbreaks are caused by a single virus variant. This is not surprising given that infected rabbit carcasses can shed viable virus for at least 20 days [72,73], making them a potent source of further infection in a population. It is therefore plausible that the first virus variant to achieve infection in a rabbit population with a sufficient density of susceptible rabbits [53] would achieve complete competitive dominance over other circulating strains in that population for that season. However, we did not detect a clear first variant to arrive in 2014 (both Clade 1 and Clade 3 were first detected on 28 August 28), and in 2013 the first RHDV1 variant detected was from Clade 2 which, despite being present in both years, was the only clade not to be found in rabbit carcasses. While the absence of Clade 2 in carcasses could be an artefact of a small sample size, the consistency of the clade found in carcasses within each site suggests that rabbit resistance to Clade 2 is more likely. That the same variants are successful in seeding an outbreak at multiple sites implies the presence of a competitive advantage impacting strain infectivity.

Our findings in this pilot study demonstrate that the presence, genetic variation and evolution of pathogens such as RHDV can be effectively examined by sequencing pooled vectors, with a fraction of the effort involved in sampling host carcasses. This method would be ideal for studying the progress of RHDV2 as it continues to spread and evolve worldwide, although more intensive methods such as capture-mark-recapture studies will also be required to assess the impact of these changes on rabbit populations.

## 4. Discussion

Passive surveillance can be useful for monitoring virus spread at a large scale when supported by a robust awareness campaign but is limited to virus detection. It is also subject to the underlying biases of data derived from the public [74]. As a complement to active monitoring, it can provide information on a virus’s geographic scale and rate of spread. The selection of appropriate surveillance techniques depends on the questions asked and resources available [75]. Fly trapping is cheaper and more sensitive than carcass hunting for detection of RHDV presence. Our results reveal that detection of pathogens through their vectors is a powerful technique for the monitoring of fly-borne diseases, unhampered by the impact of low host population density or carcass detection issues [20]. However, it does not give epidemiological data and requires laboratory expertise, while carcass searches are dependent on landscape type and scavenger activity. We have summarised the roles of different monitoring options with examples of questions they are able to address in Table 1. In many cases, a combination of different surveillance approaches will be required to obtain a full picture of disease dynamics [75].

The longevity of surveillance programs is also worth consideration. Surveillance is by nature only effective while efforts are ongoing, and projects that continue long-term enable changes to impacts of existing pathogens to be recognised and the impacts of emerging pathogens to be assessed [51]. Long-term surveillance generates a large volume of data, and potentially, biological samples both of which can yield unexpected opportunities. Our rabbit liver samples, collected through both active and passive monitoring, as well as blood sera from the Turretfield capture–mark–recapture project are stored in a frozen archive, many dating back almost 30 years. As technology has progressed, these samples have been used in an increasing variety of ways. For example, archived samples from the 1970s were used in the development of cut-off values for early RHDV antibody tests. Later, archival samples were a vital tool in the detection of a benign virus RCV-A1 that affected RHDV release programs in cooler, more temperate regions of Australia. As new ELISA tests have been developed, we have been able to return to the archival sera to distinguish between RHDV variants and between longer-term antibodies (IgG, IgA) and shorter-term IgM antibodies. This additional information has provided an increasingly clear epidemiological picture. More recently, stored capture–mark–recapture data has been used in the development of epidemiological models [52,53] while tissue samples have been accessed for population genetic studies [76]. All of these data uses, that could not have been foreseen at the beginning of the project, highlight the value of active surveillance providing long-term datasets and the importance of their careful storage.

As RHDV2/GI.2 increasingly raises concerns across the world, we hope that the South Australian RHDV monitoring experience can be useful for the design of effective surveillance globally. We encourage researchers and land managers to fully consider the scope of surveillance techniques available and select those that will best suit their needs and questions. We are mindful that passive disease surveillance will only yield useful results when community relationships are robustly invested in, and will tend to decline in interest after the hype of an initial outbreak recedes. While active monitoring requires more investment in time, money and expertise, the costs often reflect learnings that can be achieved. Active monitoring programs provide a wealth of data, the collection of which can be invaluable for training the next generation of researchers. As recently reviewed by Meletis et al. [77], it is also worth noting that being able to show freedom from disease is as important as detecting its presence.

## Figures and Tables

**Figure 1 viruses-16-01553-f001:**
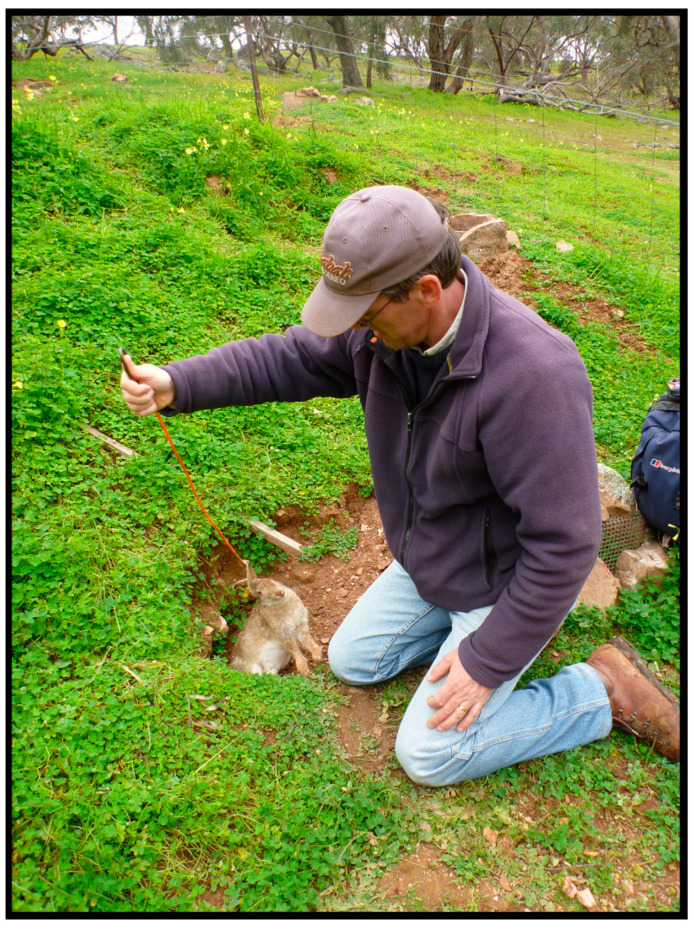
Hooking a carcass (*Oryctolagus cuniculus*) out of a burrow detected by visual checking, smelling and looking for flies moving in and out of a burrow.

**Figure 2 viruses-16-01553-f002:**
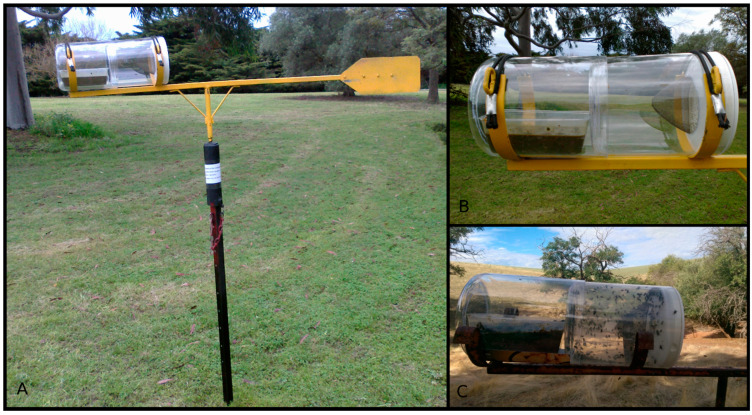
Active monitoring of rabbit haemorrhagic disease virus activity and variants by trapping of flies, the primary vectors using a wind-orienting fly trap [23]. (**A**) Full trap including base and wind-orienting vane. (**B**) Close view of trap canister with bait in the left compartment and fly-accumulating compartment on the right where flies enter through a small hole in the tip of the gauze funnel. (**C**) Close view of trap canister containing trapped flies.

**Figure 3 viruses-16-01553-f003:**
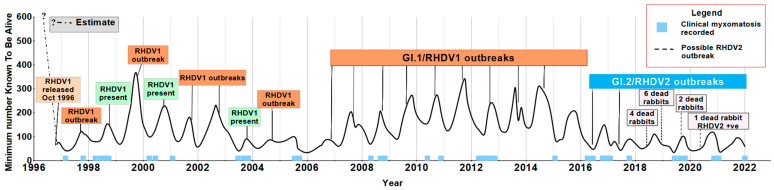
Minimum number of rabbits Known To Be Alive at Turretfield, South Australia, 1996–2022. The data was derived from live trapping approximately every eight weeks and individual tagging of captured rabbits. Estimated rabbit numbers for 1996 pre-RHDV1 release are based on ~50% of dead rabbits in 1999 outbreak being tagged, and greater pasture damage in 1996 at virus release.

**Figure 4 viruses-16-01553-f004:**
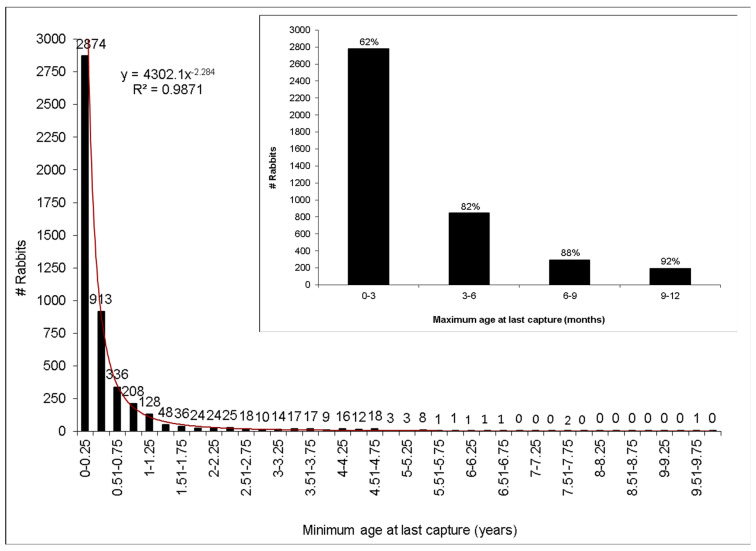
Longevity of rabbits captured at Turretfield, South Australia, 1996–2018 (*n* = 4769). Minimum age at their last capture is used because some rabbits were first captured as adults and their real age could not be estimated. Inset shows the rate of mortality for rabbits that survive < 12 months (*n* = 4118) with an initial weight ≤1300 g enabling their age to be estimated using a growth rate of 10 g/day [15; % is cumulative of total rabbits; *n* = 4455].

**Figure 5 viruses-16-01553-f005:**
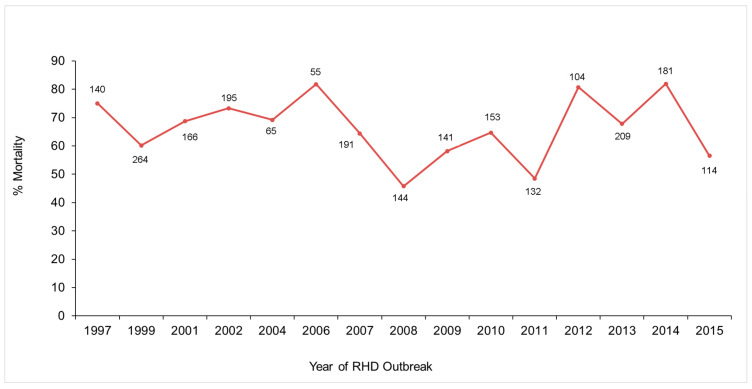
Mortality of infected rabbits at Turretfield, South Australia, following RHDV1 outbreaks (1997–2015). Data labels are the number of infected rabbits for the associated percentage mortality.

**Table 1 viruses-16-01553-t001:** Using lagomorphs as an example, a comparison of wildlife disease surveillance techniques—a summary of questions addressed, and investment required. Cells marked with “–” indicate techniques that can partially address a question.

	Passive Monitoring	Active Monitoring			
Example Questions Addressed	Community Sample Submissions	Routine Carcass Searches	Vector Sampling	Capture–Mark–Recapture	Shot Samples
Is RHDV2/GI.2 killing a lagomorph species?	✓	✓	✗	✓	✓
What genotypes are in the environment? How are they changing over time?	-	-	✓	✗	✗
Which genotypes are killing lagomorphs?	✓	✓	✗	✗	✗
What impact is a pathogen having on a population?	✗	✓	✗	✓	✗
Where and when are outbreaks occurring at a large scale?	✓	✗	✗	✗	✗
Is the mortality rate changing?	✗	-	✗	✓	✗
What proportion of a population has antibodies?	✗	✗	✗	✓	-
Which antibodies are present (e.g., RHDV1, RHDV2, benign RCV-A1)?	✗	✗	✗	✓	✓
Have lagomorphs adapted genetically to the presence of RHDV?	✗	-	✗	✓	✓
Provides material for challenge studies	✓	✓	✗	✗	✗
Relative investment required	$	$$$	$$	$$$$	$$

## Data Availability

Data associated with live-trapping studies as shown in figures are available from the author on reasonable request.

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
