# Peer review of "Surveillance of Wildlife Viruses: Insights from South Australia’s Monitoring of Rabbit Haemorrhagic Disease Virus (RHDV GI.1 and GI.2)"

_viruses, 2024, doi:10.3390/v16101553_

Round 1

Reviewer 1 Report

Comments and Suggestions for Authors

The article provides a comprehensive and informative exploration of the topic (Monitoring RHD), featuring a structure that resembles a review article despite incorporating the typical sections of a full paper. It has a technical tone, with valuable considerations, shared experiences, and practical recommendations on RHD monitoring different approaches in wild rabbit populations.

While the content is engaging and relevant, and there is nothing to be amended, to my knowledge it does not entirely align with the specific focus of this journal. For example, Figures 1 and 2, although useful, are more commonly found in technical manuals rather than scientific publications. This might reinforce the impression that the article, in its current version and language (e.g. excessive use of “we”), is better suited for a journal specializing in wildlife studies or conservation, where it could reach a more targeted audience and have a greater impact. I understand, however, that the manuscript is a direct response to the Editor's call for submissions for a special issue.

Specific comments:

Abstract

-Line 19: In addition to the USA, Mexico, and South Africa, European countries like Spain, France, and Portugal—where the conservation status of wild rabbits is a concern—also benefit from active strategies.

Key words:

-Please remove Calliphora from the keywords as it is not mentioned in the manuscript. I would suggest to include “Australia” instead.

Introduction

-Line 26-29. As written in lines 26 to 29, the authors seem to suggest that the viruses mentioned, namely ASFV and RHDV could become zoonotic. However, in the cited references, such as reference 3 for ASFV, there is no mention of this potential. I would suggest replacing the references. In fact, many swine viruses have been considered potentially zoonotic (F. Ruiz-Fons, 2017), but, to my knowledge, ASFV is not one of them.

 -Line 33. “other researchers” or “researches”?

-I suggest that at the end of the introduction section, the authors add a short paragraph explaining that this review article aims to provide a summary of the advantages and disadvantages of 5 types of RHD monitoring and share experiences with the scientific community.

Materials and Methods

-Line 108. When the authors mention that the necropsies were performed at the location where the carcasses were collected, they should also state what measures were taken to prevent local contamination.

-Line 111. Please provide a reference for the use of size of the maggots for estimate the date of death.

Results:

-Section 3.5. First paragraph. Please provide a reference for the PCR methods mentioned.

Discussion:

Table 1.

-Please explain the meaning of the dash (-).

-Please correct “Shot transects”.

-Please amend “Investment Required” to “Relative investment required” since no values are presented.

Figures:

-Use the same type of font in all figures.

-Figure 1 and 2, should you decided to keep, must be standardized; they should either both have a black border or neither should have one.

-Figure 3. the font size in the x-axis legend is so small that cannot be read.

Author Response

Reviewer 1

The article provides a comprehensive and informative exploration of the topic (Monitoring RHD), featuring a structure that resembles a review article despite incorporating the typical sections of a full paper. It has a technical tone, with valuable considerations, shared experiences, and practical recommendations on RHD monitoring different approaches in wild rabbit populations.

While the content is engaging and relevant, and there is nothing to be amended, to my knowledge it does not entirely align with the specific focus of this journal. For example, Figures 1 and 2, although useful, are more commonly found in technical manuals rather than scientific publications. This might reinforce the impression that the article, in its current version and language (e.g. excessive use of “we”), is better suited for a journal specializing in wildlife studies or conservation, where it could reach a more targeted audience and have a greater impact. I understand, however, that the manuscript is a direct response to the Editor's call for submissions for a special issue.

Yes, this submission is in response to a direct request from the guest editor of this special issue who saw need for our experiences to be shared with the international community. We recognise that our article is an unusual combination of review and novel data, which doesn’t neatly fit within the journal’s usual formats, but see it as a valuable contribution closely aligned to this special edition.

Specific comments:

Abstract

-Line 19: In addition to the USA, Mexico, and South Africa, European countries like Spain, France, and Portugal—where the conservation status of wild rabbits is a concern—also benefit from active strategies.

True. Added.

Key words:

-Please remove Calliphora from the keywords as it is not mentioned in the manuscript. I would suggest to include “Australia” instead.

Done

Introduction

-Line 26-29. As written in lines 26 to 29, the authors seem to suggest that the viruses mentioned, namely ASFV and RHDV could become zoonotic. However, in the cited references, such as reference 3 for ASFV, there is no mention of this potential. I would suggest replacing the references. In fact, many swine viruses have been considered potentially zoonotic (F. Ruiz-Fons, 2017), but, to my knowledge, ASFV is not one of them.

Our examples were of “Monitoring the evolution and spread of viral diseases”. The point about becoming zoonotic was additional. Additional words added to following sentence to clarify the viral disease examples were about “Monitoring the evolution and spread of viral diseases”, not them potentially becoming zoonotic.

-Line 33. “other researchers” or “researches”? Researchers is correct

-I suggest that at the end of the introduction section, the authors add a short paragraph explaining that this review article aims to provide a summary of the advantages and disadvantages of 5 types of RHD monitoring and share experiences with the scientific community.

Added: In this article we share our experience with five different wildlife virus surveillance techniques and summarise their relative advantages and disadvantages to facilitate the development of considered monitoring strategies.

Materials and Methods

-Line 108. When the authors mention that the necropsies were performed at the location where the carcasses were collected, they should also state what measures were taken to prevent local contamination.

Added: New gloves and scalpel are used for each carcass to minimise the risk of cross contamination. Carcasses are left where they were found to continue spreading the virus naturally.

-Line 111. Please provide a reference for the use of size of the maggots for estimate the date of death.

Reference added.

Results:

-Section 3.5. First paragraph. Please provide a reference for the PCR methods mentioned.

We have now added the primers and their citation to the Methods section. Full details of genetic methods are still provided in the supplementary materials.

Discussion:

Table 1.

-Please explain the meaning of the dash (-).

The table caption specifies that “cells marked with “–” indicate techniques that can partially address a question.”

-Please correct “Shot transects”.

Changed to Shot samples to match methods heading

-Please amend “Investment Required” to “Relative investment required” since no values are presented.

Done

Figures:

-Use the same type of font in all figures.

Fixed

-Figure 1 and 2, should you decided to keep, must be standardized; they should either both have a black border or neither should have one.

Similar border added to Figure 1. Figure 2 requires the internal borders to separate the three images.

-Figure 3. the font size in the x-axis legend is so small that cannot be read.

Fixed

Reviewer 2 Report

Comments and Suggestions for Authors

Manuscript ID: viruses-3220861

 Surveillance of wildlife viruses: insights from South Australia’s monitoring of rabbit haemorrhagic disease virus (RHDV GI.1 and GI.2)

 This paper reviews and discusses the experience acquired for over 30 years of surveillance of virus circulation (RHDV GI.1 and GI.2) in wild rabbit populations (Oryctolagus cuniculus) in South Australia. It takes into account both, passive and active surveillance approaches, considering their strengths and weaknesses, concerning cost and knowledge provided. Gained insights are discussed.

 I think the paper provides interesting information, which is especially relevant in the context of the current situation caused by RHDV2 spreading through Europe, North America, South Africa and Asian countries, affecting a large number of susceptible wildlife lagomorphs, including endangered species. The South Australian RHDV monitoring experience can provide insights relevant to gaining a better understanding of the impacts of RHDV2 and other wildlife pathogen monitoring systems globally. I found no major issues, I only have minor comments.

 Minor comments:

 - The line numbers disappear after page 6, making difficult the review process.

 - Paragraph included in lines 239-248 (Results section 3.2), regarding capture-mark-recapture studies, summarises in a very straightforward way the results achieved in a long-term study that has not been previously introduced in this section. It is true that some information is given previously in Materials and Methods section 2.2, but I think the introduction of the long-term study can be improved to facilitate the readers' understanding. For example, the date of the first official release of RHDV at Turretfield is not indicated (was it in 1996, short after the official authorisation from the authorities? or several years after). Regarding the research site, readers are referred to reference 14, but maybe a brief outline could be given.

 - Figure 3. The text labels are not visible to the human eye.

- First line of page 8. “the trappable portion of the population”. Maybe a brief explanation could be given if there are some known factors affecting trapability of rabbits.

 - The issue about the change in behaviour in adult rabbits causing a dropped trappability, concomitant to the spread of RHDV2 sounds quite interesting. Is there any speculation about its cause? It has been a transitory effect or lasts until today?

 - Section 3.5 Vector surveys. Authors indicate flies were captured using fly traps and in parallel rabbit carcasses were searched in the surroundings, but, it is not clear if flies were also collected from carcasses.

 - Last paragraph of section 3.5: “(16 recombination events were detected affecting 51 contigs of 85 total)” is there a reference for this?

 - Maybe in this paragraph regarding the relevance of recombination events in Lagoviruses evolution it could be mentioned that, in fact, all currently circulating RHDV2 viruses seem to have emerged from recombination events involving different Lagoviruses, as indicated in:

 Abrantes, J.; Droillard, C.; Lopes, A.M.; Lemaitre, E.; Lucas, P.; Blanchard, Y.; Marchandeau, S.; Esteves, P.J.; Le Gall-Recule, 818 G. Recombination at the emergence of the pathogenic rabbit haemorrhagic disease virus Lagovirus europaeus/GI.2. 819 Scientific Reports 2020, 10, doi:10.1038/s41598-020-71303-4.

Author Response

Reviewer 2

This paper reviews and discusses the experience acquired for over 30 years of surveillance of virus circulation (RHDV GI.1 and GI.2) in wild rabbit populations (Oryctolagus cuniculus) in South Australia. It takes into account both, passive and active surveillance approaches, considering their strengths and weaknesses, concerning cost and knowledge provided. Gained insights are discussed.

 I think the paper provides interesting information, which is especially relevant in the context of the current situation caused by RHDV2 spreading through Europe, North America, South Africa and Asian countries, affecting a large number of susceptible wildlife lagomorphs, including endangered species. The South Australian RHDV monitoring experience can provide insights relevant to gaining a better understanding of the impacts of RHDV2 and other wildlife pathogen monitoring systems globally. I found no major issues, I only have minor comments.

 Minor comments:

 - The line numbers disappear after page 6, making difficult the review process.

Fixed

 - Paragraph included in lines 239-248 (Results section 3.2), regarding capture-mark-recapture studies, summarises in a very straightforward way the results achieved in a long-term study that has not been previously introduced in this section. It is true that some information is given previously in Materials and Methods section 2.2, but I think the introduction of the long-term study can be improved to facilitate the readers' understanding. For example, the date of the first official release of RHDV at Turretfield is not indicated (was it in 1996, short after the official authorisation from the authorities? or several years after). Regarding the research site, readers are referred to reference 14, but maybe a brief outline could be given.

Details of the official release of RHDV and the subsequent monitoring program are provided on lines 91-100. We believe this provides sufficient details.

 - Figure 3. The text labels are not visible to the human eye.

Fixed as for Reviewer 1 above

- First line of page 8. “the trappable portion of the population”. Maybe a brief explanation could be given if there are some known factors affecting trapability of rabbits.

 - The issue about the change in behaviour in adult rabbits causing a dropped trappability, concomitant to the spread of RHDV2 sounds quite interesting. Is there any speculation about its cause? It has been a transitory effect or lasts until today?

It is a standard premise of any activity that it only impacts individuals of the population that are susceptible. Hence ‘integrated pest management’ utilises a number of control tools to ensure the maximum proportion of the population is controlled. Rabbit cage trapping is no different and though it attempts to trap all individuals, it is highly likely that some rabbits are averse to the traps and/or bait and hence minimally, if at all, trappable. See for example Daly, J. (1980).  Age, sex and season: Factors which determine the trap response of the European wild rabbit, Oryctolagus cuniculusWildlife Research 7, 421-432. We speculate that the 80% reduction in the Turretfield rabbit population from RHDV2/GI.2 (Mutze et al. 2018) caused social impacts, including heightened nervousness and neophobia, or increased food availability and hence less attractiveness of the carrot bait, in many adult rabbits. How long this apparent reduced trappability of adult rabbits continued is unknown as the experienced researchers Sinclair and Peacock ended their work at Turretfield in 2018.

The words “a standard premise of all such activities,” have been added.

 - Section 3.5 Vector surveys. Authors indicate flies were captured using fly traps and in parallel rabbit carcasses were searched in the surroundings, but, it is not clear if flies were also collected from carcasses.

Added: No attempt was made to collect flies directly from carcasses.

 - Last paragraph of section 3.5: “(16 recombination events were detected affecting 51 contigs of 85 total)” is there a reference for this?

No reference as this is our observed result. The detailed methods in the supplementary materials contain relevant references for the methods we used.

 - Maybe in this paragraph regarding the relevance of recombination events in Lagoviruses evolution it could be mentioned that, in fact, all currently circulating RHDV2 viruses seem to have emerged from recombination events involving different Lagoviruses, as indicated in:

Abrantes, J.; Droillard, C.; Lopes, A.M.; Lemaitre, E.; Lucas, P.; Blanchard, Y.; Marchandeau, S.; Esteves, P.J.; Le Gall-Recule, 818 G. Recombination at the emergence of the pathogenic rabbit haemorrhagic disease virus Lagovirus europaeus/GI.2. 819 Scientific Reports 2020, 10, doi:10.1038/s41598-020-71303-4.

Added: Indeed, the RHDV2 genotype (GI.2) which is currently dominant worldwide appears to be a product of recombination which has resulted in increased infectivity in young rabbits and hares (Abrantes et al 2020).

Reviewer 3

"One thought on this manuscript - the science of health surveillance has really developed over the last 15 years and it's important for current approaches to be reflected in an article like this.
The following links provide useful background to this:
https://www.woah.org/fileadmin/Home/eng/Internationa_Standard_Setting/docs/pdf/A_TAHSC_Feb_2018_Part_B.pdf
https://www.frontiersin.org/journals/veterinary-science/articles/10.3389/fvets.2020.637364/full
https://pubmed.ncbi.nlm.nih.gov/31731037/     "

We thank this reviewer for providing these links. Four additional references and some additional text have been added to the manuscript. All changes and reference additions are highlighted in yellow.